# Prevalence and correlates of voluntary medical male circumcision adverse events among adult males in the Copperbelt Province of Zambia: A cross-sectional study

Imukusi Mutanekelwa[1], Seter Siziya[1], Victor Daka[1,2]*, Elijah Kabelenga[3], Ruth L. Mfune[1,2], Misheck Chileshe[4], David Mulenga[1], Herbert Tato Nyirenda[1], Christopher Nyirenda[1], Steward Mudenda[2,5], Bright Mukanga[1], Kasonde Bowa[6]

1 Michael Chilufya Sata School of Medicine, Copperbelt University, Ndola, Zambia, 2 School of Veterinary Medicine, University of Zambia, Lusaka, Zambia, 3 Ndola Teaching Hospital, Ndola, Zambia, 4 Marybegg Health Services, Ndola, Zambia, 5 School of Health Sciences, University of Zambia, Lusaka, Zambia, 6 University of Lusaka, Lusaka, Zambia

* dakavictorm@gmail.com

**Data Availability Statement:** All relevant data are within the paper and its Supporting Information files.

## Abstract

### Background

Voluntary Medical Male Circumcision (VMMC) is a key intervention in HIV/AIDS. Improving VMMC program uptake in Zambia requires careful monitoring of adverse events (AE) to inform program quality and safety. We investigate the prevalence of VMMC AE and their associated factors among adult males in Ndola, Copperbelt Province, Zambia.

### Methods

We performed a cross-sectional study using secondary clinical data collected in 2015 using two validated World Health Organisation/Ministry of Health reporting forms. We reviewed demographics and VMMC surgical details from 391 randomly sampled adult males aged ≥18 years at Ndola Teaching Hospital, a specialised VMMC fixed site in Zambia. Non-parametric tests (Fisher's exact test or Chi-square depending on assumptions being met) and logistic regression were conducted to determine the relationships between associated factors and VMMC AE.

### Results

The overall VMMC AE prevalence was 3.1% (95% CI 1.60%– 5.30%) and most AEs occurred postoperatively. In decreasing order, the commonly reported VMMC AE included; bleeding (47.1%), swelling (29.4%), haematoma (17.6%), and delayed wound healing (5.9%). There was an inversely proportional relationship between VMMC volume (as measured by the number of surgeries conducted per VMMC provider) and AEs. Compared to the highest VMMC volume of 63.2% (247/391) as reference, as VMMC volume reduced to 35.0% (137/391) and then 1.8% (7/391), the likelihood of AEs increased by five times (aOR

**Funding:** The author(s) did not receive any funding for this work.

**Competing interests:** The authors declare no competing interests exist.

5.08; 95% CI 1.33–19.49; p = 0.018) and then sixteen times (aOR 16.13; 95% CI 1.42–183.30; p = 0.025) respectively.

## Conclusions

Our study found a low prevalence of VMMC AEs in Ndola city, Copperbelt Province of Zambia guaranteeing the safety of the VMMC program. We recommend more surgically proficient staff to continue rendering this service. There is a need to explore other high priority national/regional areas of VMMC program safety/quality, such as adherence to follow-up visits.

## Introduction

Voluntary Medical Male Circumcision (VMMC) is a key intervention in HIV/AIDS [1]. Since 2007, VMMC has been recommended by WHO/UNAIDS as efficacious for the prevention of HIV in a country such as Zambia which has heterosexual epidemics, low VMMC, and high HIV prevalence [1, 2]. Evidence from randomised clinical trials (RCT) have shown VMMC to provide 60% partial protection by reducing the risk of acquiring HIV in heterosexual men [3–5] with the level of protection increasing to 74% over time [6]. In contrast to Northern Africa where traditional circumcision is practiced extensively, VMMC practice is rare in Sub-Saharan Africa (SSA), an area disproportionately affected by HIV/AIDS [7]. For instance, Zambia still has one of the highest HIV prevalence rates in SSA [8].

The World Health Organisation (WHO) recommends the scale-up of VMMC in low Male Circumcision and high HIV prevalence countries of SSA with careful monitoring of adverse events which are not to exceed 5% [9]. However, the reported VMMC adverse events (AE) estimates range from 1.0–6% [3–5, 10–13] with the lowest AE rates seen intraoperatively with a prevalence between 0.1% - 0.2% [10, 14]. In ascending order, the prevalence of VMMC adverse event in various SSA countries was as follows: 1.0% South Africa [3], 1.5% Kenya [4], 2.0 Kenya [11], 2.2% Kenya [10], 3.6% Uganda [5], 5.1% Kenya [13], and 5.47% Uganda [12]. One of the main WHO monitoring and evaluation parameters is the number or percentage of circumcised males experiencing an adverse event during and after the VMMC procedure [15]. This parameter is important because it provides information to determine the overall safety/ quality of a VMMC program, which also influences an individual's perceived safety concerns and ultimately affects VMMC coverage [15]. According to the Zambia National VMMC operational plan 2016–2020, studies concerning VMMC adverse events is a research priority at the national and regional level to improve the quality of the VMMC program [16]. There has been paucity of data and little published work on the demand-side factors affecting the VMMC coverage such as safety/quality of the VMMC and adverse events in Zambia. The current study attempts to redress this critical knowledge gap. The objective of the study was to determine the prevalence of VMMC adverse events among adult males in Ndola on the Copperbelt Province of Zambia using a cross-sectional study design.

## Materials and methods

### Study design

We conducted a cross-sectional study using secondary clinical data collected from January to December 2015 at Ndola Teaching Hospital in Copperbelt Province of Zambia. Ndola

Teaching Hospital is a third level specialised referral healthcare facility which is the largest in the northern region of Zambia [17]. The hospital is a fixed VMMC site and during school holidays adopts a campaign VMMC model. A VMMC campaign model involves actively mobilising pupils and conducting VMMC during the school holidays to avoid disruption of learning time [16]. At the hospital, conventional surgery is employed for VMMC as medical devices are not stocked nor used. Though conventional surgery can be done by one of three methods (Dorsal Slit, Sleeve, or Forceps Guided), WHO recommends forceps guided technique to be used [9]. The hospital is responsible for the seven core VMMC responsibilities, serves as a clinical training hub for the lower levels, and is a referral centre when AEs need specialised attention. The seven core components of VMMC include registration/waiting, group education, individual HIV testing/counselling, clinical screening, VMMC surgery, immediate postoperative care, and lastly postoperative follow up care.

## Sample size estimation

The sample size estimation was based on the prevalence of VMMC AE of 8% obtained from a previous study done in neighbouring Malawi because of similarities in VMMC training and reporting guidelines [18]. Employing a precision of 3% and accounting for a 20% missing clinical information, a minimum sample size of 377 adult males was determined. However, a total of 391 records which met the inclusion criteria were available and included in the study. The inclusion criteria were adult males aged 18 years and above who underwent VMMC at Ndola Teaching Hospital in 2015, while males less than 18 years were excluded from the study.

## Sources of secondary data

This study used secondary VMMC clinical data collected at Ndola Teaching Hospital from January to December 2015. The hospital recorded the VMMC details and outcomes using two WHO/Ministry of Health validated reporting forms which researchers accessed to collect the data. The first form accessed by researchers was called the 'VMMC client record form' and data were collected for each adult male sampled. Researchers reviewed the second form called the 'VMMC Adverse Events form' for the few adult males who experienced adverse events as this form is not available when an adverse event has not occurred. The 'VMMC client record form' provided information such as sociodemographic characteristics, history taken, physical examination details, investigations conducted, surgical operation details (pre-op preparation, procedure type, duration, provider, outcome), and follow-up visits. If an adverse event occurred, the VMMC provider initially records it in the 'VMMC client record form' however with fewer details as most details are made available in the 'VMMC Adverse Events form'. The 'VMMC Adverse Events form', is only used when an adverse event has occurred, and it provides detailed information regarding the type of adverse event, severity (mild, moderate or severe), treatment provided/ outcome, and the timing of the adverse event [15]. The types of VMMC adverse events in the latter form includes pain, infection, delayed wound healing, excessive bleeding, swelling/haematoma, insufficient or excessive skin removed, damage to the penis, appearance, and voiding problems.

## Ethical considerations

The Biomedical Research Ethics Committee (#BE064/16) at the University of Kwazulu-Natal in South Africa approved this study. The Ndola Teaching Hospital management granted access to review clinical records. Confidentiality was maintained by not publishing unique identifiers and restricting data accessibility to the research team only.

## Statistical analysis

All data were entered and analysed using Statistical Package for the Social Sciences (SPSS) v.25. The computed descriptive statistics were expressed as frequencies and tabulated. The prevalence of adverse VMMC AE was calculated with its 95% CI using the Exact method by Clopper and Pearson (1934). The relationships between predictor variables and AE were assessed using Fisher's exact test when the test assumption of expected frequency for each cell to be at least five was not met for the use of Pearson chi-square ($\chi 2$). Binomial logistic regression using a stepwise backward elimination (likelihood ratio) method for selection of variables was employed to control for all confounders. For inclusion in the multivariate logistic regression model, variables needed a p $\leq$0.05 in the univariate binomial logistic regression. For both analyses, a result yielding p value < 0.05 was regarded as statistically significant. Incompletely filled in VMMC forms led to variables with missing values; missing values were handled using listwise deletion by excluding them from the analysis [19].

## Results

### Adverse events and sociodemographic characteristics

We reviewed a total of 391 VMMC records of participants from Ndola Teaching Hospital. The estimated overall prevalence for VMMC AE was 3.1% (95% CI 1.60%– 5.30%) in Ndola city, Copperbelt Province of Zambia. The prevalence was higher (10/391, 2.6%) in the postoperative period (see Fig 1). The commonly reported VMMC AEs during the 48 hours, 7 days and 4

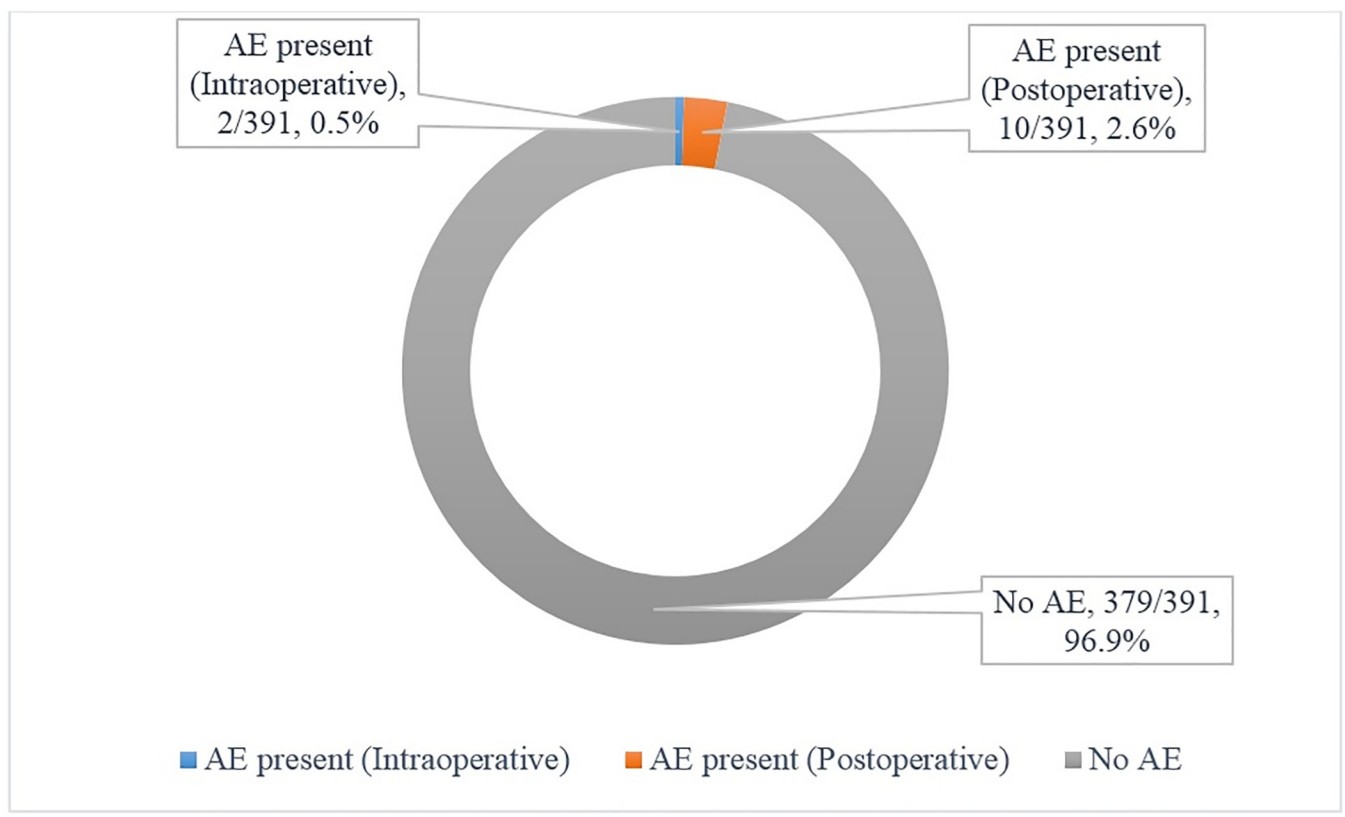

**Fig 1. Prevalence and timing of VMMC AE among adult males.**

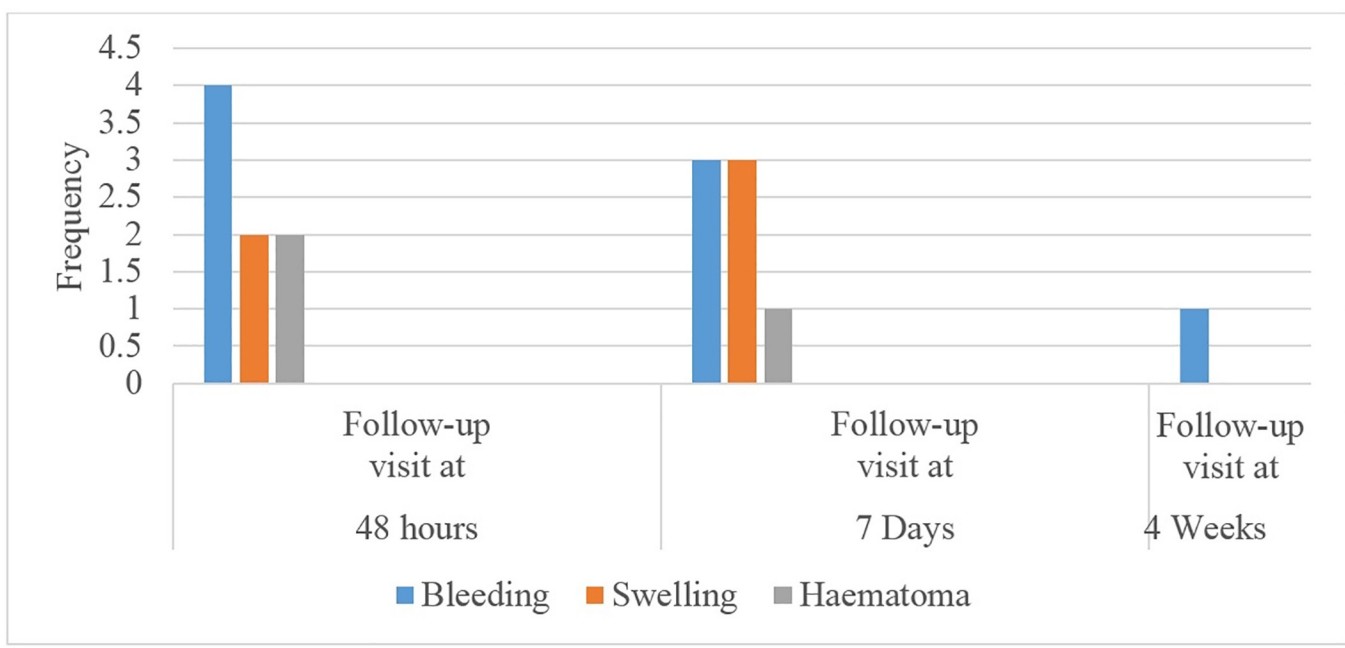

**Fig 2. Frequency and types of AE experienced during the follow-up visits.**

weeks postoperative follow-up visit were bleeding (4/8), infection (3/7), and delayed wound healing (1) respectively (see Fig 2).

There were higher proportions of AEs noted in the age group 50+ years (1/14, 7.1%) compared to the younger age group (11/377, 2.9%. The AE proportion was similar for those who were married (4/123, 3.3%) and those who were not (8/263, 3.0%). Individuals who indicated the reason for undergoing VMMC as partial protection against acquiring HIV experienced similar AE proportions (10/308, 3.3%) compared to those with other reasons (1/45, 2.2%). Furthermore, using Fisher's exact test, reasons for undergoing VMMC and sociodemographic characteristics were not significantly associated with AEs (see Table 1).

**Table 1. Prevalence of AE according to sociodemographic characteristics.**

| | Total, n (%)* | No Adverse Event | Adverse Event present | P value |
|---|---|---|---|---|
| Ages | | | | |
| 18–49 | 376 (100) | 365 (97.1) | 11 (2.9) | 0.359 |
| 50+ | 14 (100)** | 13 (92.9) | 1 (7.1) | |
| Marital Status | | | | |
| Not Married | 263 (100) | 255 (97.0) | 8 (3.0) | 1.000 |
| Married | 123 (100) | 119 (96.7) | 4 (3.3) | |
| Tribe | | | | |
| Bemba | 139 (100) | 138 (99.3) | 1 (0.7) | 0.063 |
| Non-Bemba | 246 (100) | 235 (99.5) | 11 (4.5) | |
| Primary Indication for MC | | | | |
| Partial Protection against HIV | 308 (100) | 298 (96.8) | 10 (3.2) | 1.000 |
| Other | 45 (100) | 44 (97.8) | 1 (2.2) | |

* Row percents.

** Interpret the percentages with caution since the denominator is less than 30.

**Table 2. Prevalence of AE according to surgical operation details.**

| | Total, n (%)* | No Adverse Event | Adverse Event present | P value |
|---|---|---|---|---|
| VMMC Volume$^{\pm}$, n = 391 | | | | |
| **Enrolled Nurses (ZEN, ETN)** | **247 (100)** | **244 (98.8)** | **3 (1.2)** | **0.008** |
| **Registered Nurses (RN, RTN, RM)** | **137 (100)** | **129 (94.2)** | **8 (5.8)** | |
| **Medical Officer (JRMO, Surgeon)** | **7 (100)\*\*** | **6 (85.7)** | **1 (14.3)** | |
| Date of operation, n = 391 | | | | |
| January–June | 187 (100) | 182 (97.3) | 5 (2.7) | 0.664\*\*\* |
| July–December | 204 (100) | 197 (96.6) | 7 (3.4) | |
| Duration, n = 347 | | | | |
| < 20 minutes | 96 (100) | 93 (96.9) | 3 (3.1) | 0.834 |
| 20–30 minutes | 185 (100) | 179 (96.8) | 6 (3.2) | |
| > 30 minutes | 66 (100) | 65 (98.5) | 1 (1.5) | |
| Local Anaesthetic amount used, n = 236 | | | | 0.604 |
| Between 8–10 ml | 188 (100) | 184 (97.9) | 4 (2.1) | |
| Below 8 or above 10 | 48 (100) | 46 (95.8) | 2 (4.2) | |
| VMMC Type, n = 386 | | | | |
| Dorsal Slit | 384 (100) | 373 (97.1) | 11 (2.9) | 0.061 |
| Sleeve Resection | 2 (100)* | 1 (50.0) | 1 (50.0) | |
| Follow-up immediately post op, n = 391 | | | | |
| Yes | 386 (100) | 375 (82.4) | 11 (2.8) | 0.145 |
| No | 5 (100)* | 4 (80.0) | 1 (20.0) | |
| Follow-up visit at 48 hours, n = 391 | | | | |
| Yes | 333 (100) | 322 (96.7) | 11 (3.3) | 1.000 |
| No | 58 (100) | 57 (98.3) | 1 (1.7) | |
| Follow-up visit at 7 days, n = 391 | | | | |
| Yes | 216 (100) | 209 (96.8) | 7 (3.2) | 0.827\*\*\* |
| No | 175 (100) | 170 (97.1) | 5 (2.9) | |
| Follow-up visit at 4 weeks, n = 391 | | | | |
| Yes | 61 (100) | 57 (93.4) | 4 (6.6) | 0.100 |
| No | 330 (100) | 322 (97.6) | 8 (2.4) | |
| Follow-up at least 1 visit post VMMC, n = 391 | | | | |
| Yes | 350 (100) | 339 (96.9) | 11 (3.1) | 1.000 |
| No | 41 (100) | 40 (97.6) | 1 (2.4) | |

$\pm$ ZEN = Zambia Enrolled Nurse, ETN = Enrolled Theatre Nurse, RN = Registered General Nurse, RTN = Registered Theatre Nurse, RM = Registered Midwife, JRMO = Junior Resident Medical Officer.

\* Row percents.

\*\* Interpret the percentages with caution since the denominator is less than 30.

\*\*\* Chi-square was valid.

## VMMC operation details

From Table 2, regarding the VMMC volume as measured by the number of circumcisions conducted, two individual enrolled nurses completed more than half (247/391, 63.2%) followed by three individual registered nurses (137/391, 35.0%) and lastly two individual medical officers (7/391, 1.8%). The VMMC AE proportion in medical officers was one in seven (1/7, 14.3%) and for nurses (11/384, 2.9%). There were similar AE proportions noted for VMMC conducted in the first semester (5/187, 2.7%) and second semester (7/204, 3.4%) of 2015.

**Table 3. Factors associated with VMMC AE after univariate binomial regression.**

| | OR (95% CI) | P value |
|---|---|---|
| Age (years), n = 390 | | 0.386 |
| 18–49 years | 1 | |
| 50+ years | 2.55 (0.31–21.2) | |
| Marital Status, n = 386 | | 0.912 |
| Not Married | 1 | |
| Married | 1.07 (0.32–3.63) | |
| Tribe, n = 385 | | 0.076 |
| Bemba | 1 | |
| Non-Bemba | 6.46 (0.825–50.57) | |
| Primary Indication for VMMC, n = 353 | | 0.713 |
| Partial Protection against acquiring HIV | 1 | |
| Other | 0.68 (0.09–5.42) | |
| **VMMC Volume, n = 391** | | |
| **Enrolled Nurses (ZEN, ETN)** | **1** | **0.027** |
| **Registered Nurses (RN, RTN, RM)** | **5.04 (1.32–19.34)** | **0.018** |
| **Medical Officer (JRMO, Surgeon)** | **13.56 (1.23–149.98)** | **0.034** |
| Date of operation via Quarter, n = 391 | | 0.665 |
| January–June | 1 | |
| July–December | 1.29 (0.40–4.15) | |
| Duration, n = 347 | | |
| < 20 minutes | 1 | 0.770 |
| 20–30 minutes | 1.04 (0.25–4.25) | 0.957 |
| > 30 minutes | 0.48 (0.05–4.69) | 0.525 |
| Local Anaesthetic amount used, n = 236 | | 0.432 |
| Between 8–10 ml | 1 | |
| Below 8 or above 10 | 2.00 (0.36–11.26) | |
| **VMMC Type, n = 386** | | **0.015** |
| **Dorsal Slit** | **1** | |
| **Sleeve Resection** | 33.91 (1.99–578.02) | |
| Follow-up visit at 48 hours, n = 391 | | 0.527 |
| Yes | 1 | |
| No | 0.51 (0.07–4.06) | |
| Follow-up visit at 7 days, n = 391 | | 0.827 |
| Yes | 1 | |
| No | 0.88 (0.27–2.82) | |
| Follow-up visit at 4 weeks, n = 391 | | 0.099 |
| Yes | 1 | |
| No | 0.35 (0.10–1.22) | |
| Follow-up at least 1 visit post VMMC, n = 391 | | 0.805 |
| Yes | 1 | |
| No | 0.77 (0.10–6.13) | |

There were variable VMMC AE proportions according to the duration of the surgery, volume of lignocaine used, VMMC type and males who attended VMMC follow-up visit. Using Fisher's exact test, only one variable, namely VMMC volume (p = 0.008) was significantly associated with AE while all other VMMC operation factors were not (see Table 2).

**Table 4. Variables included in the multivariate analysis.**

| | Multivariate Analysis | |
|---|---|---|
| | aOR (95% CI) | P value |
| **VMMC Volume, n = 391** | | |
| Enrolled Nurses (ZEN, ETN) | 1 | 0.022 |
| Registered Nurses (RN, RTN, RM) | 5.08 (1.33–19.49) | 0.018 |
| Medical Officer (JRMO, Surgeon) | 16.13 (1.42–183.30) | 0.025 |

## Factors associated with VMMC AE

Table 3 depicts the results of the univariate binomial regression analysis which shows the association between AE and various independent variables. Independent variables included socio-demographic factors and VMMC pre and post operation details. Only two factors were univariately associated with AEs and these were VMMC volume (as measured by the number of surgeries conducted per VMMC provider) and VMMC type. Compared to the highest VMMC volume (63.2%) as reference, as VMMC volume reduced to 35.0% and then 1.8% the likelihood of AEs increased by five times (OR 5.044; 95% CI 1.316–19.339; p = 0.018) and then thirteen times (OR13.556; 95% CI 1.225–149.984; p = 0.034) respectively. Regarding surgical technique used, AEs occurred 97% less when Sleeve Resection was used relative to Dorsal Slit (OR 0.029; 95% CI 0.002–0.503; p = 0.038). The rest of the factors were not associated with AE, see Table 3 for more details.

To control for confounders, multivariate binomial regression was used to model the association between AE and selected variables at p ≤ 0.05 which included VMMC volume and VMMC type (see Table 4). The regression model significantly ($\chi^2$ 8.41, p = 0.015) accounted for the factors associated with AEs and this represented up to 8.9% of the factors affecting AEs (based on Nagelkerke $R^2$). VMMC volume was the only factor significant and independently associated with adverse events. There was an inversely proportional relationship between VMMC volume and AEs as lower volumes were associated with more AEs. Compared to the highest VMMC volume of 63.2% (247/391), the odds of experiencing an adult VMMC AE was significantly increased by five times (aOR 5.08; 95% CI 1.33–19.49; p = 0.018) with a VMMC volume of 35.0% (137/391) and sixteen times (aOR 16.13; 95% CI 1.42–183.30; p = 0.025) with a VMMC volume of 1.8% (7/391) (see Table 4). In decreasing order of VMMC volume, enrolled nurses conducted the most (63.2%) followed by registered nurses (35.0%) and lastly medical officers (1.8%).

## Discussion

Our study investigated the specific demand-side factors affecting the low VMMC coverage by focusing on the safety/quality of the VMMC program in Zambia. This was done by estimating the prevalence of VMMC adverse events among adult males and assessing associated factors in this group in Ndola city of Copperbelt Province, Zambia. We found that the estimated prevalence of VMMC AEs among adult males in Ndola city, Copperbelt Province was generally low (3.1%) and tended to be higher in the postoperative (2.6%) compared to the intraoperative period (0.5%). The prevalence of VMMC AEs in the intraoperative period of our study was slightly higher than the two prospective cohort studies done in Kenya and South Africa[10, 14]. This was probably attributed to VMMC providers with less surgical proficiency (i.e. Junior Resident Medical Officer) and the use of surgically advanced VMMC methods (i.e. Sleeve Resection) in less experienced clinician hands; this result must be interpreted with extreme caution because of the few numbers of Sleeve Resections done. To reduce AEs, WHO

recommends the use Dorsal Slit as the surgical method of choice because of its safety, needing less theatre assistance, requiring less surgical expertise, easy to learn and gain proficiency [18]. Evidence from other cross-sectional studies have shown that the overall prevalence of VMMC AE is between 0.2% to 8.6%% which resonates with our studies results [15–17]. Besides, more robust studies (i.e., prospective cohorts) in SSA have also established that the prevalence of VMMC AE ranged from 2.0% to 5.5% which is also consistent with our findings [10–13]. The dissimilarity in prevalence may be caused by variability in sample sizes and study types (experimental versus observational).

Our study showed that there was a significant association between AE and the VMMC volume (as measured by the number of surgeries conducted per VMMC provider). There was an inversely proportional relationship between AE and VMMC volume because the more a provider performed circumcisions, experience increased and thus the less likely an adverse event would occur, the converse was also correct. In our study, most circumcisions were performed by enrolled nurses, then registered nurses and lastly medical officers. This study did not focus on the quality of surgical techniques necessary to compare AEs among the VMMC providers using a prospective cohort study. The AEs differences among VMMC providers is multifactorial and inclusive of differences in VMMC volume. All VMMC providers undergo standardised training and before being certified, must demonstrate surgical competence by performing a specified number of circumcisions under close supervision [9, 16]. However, to maintain the desired level of competence to gain surgical proficiency, the routine practice, especially after certification is recommended [9]. Despite all VMMC providers in our study being certified surgically competent, the surgically proficiency was questionable with very low frequency and number of VMMC conducted. Hence the differences in AE are probably because enrolled nurses gained the most surgical proficiency as they performed the majority of circumcisions (63.2%), followed by registered nurses (35.0%) and lastly medical officers (1.8%). These findings are similar to a prospective cohort study in Kenya which showed that VMMC providers (nurses and clinicians) who performed a large number of circumcisions gained surgical proficiency and were significantly less likely to cause adverse events [10].

There were differences in the types of VMMC AE experienced among studies during the postoperative follow-up visit. In our study, the frequency of commonly reported VMMC AEs postoperatively at the 48 hours follow-up visit were bleeding (4/8), swelling (2/8), and haematoma (2/8). Our findings are similar to a Zimbabwean cross-sectional study which reported that bleeding was the most common VMMC AE in the first 48 hours [20]. During the 7 days follow-up visit, the frequency of reported VMMC AEs were bleeding (3/7), swelling (3/7), and haematoma (1/7) in our study. In comparison, a South African prospective cohort study showed that swelling and wound infection were the most common VMMC AEs at 7 days [14]. In contrast to our study, a Malawian study reported more VMMC AEs which included in decreasing order infections, haematoma, swelling, delayed wound healing, bleeding, and wound disruption [15] while a retrospective case series analysis done in Tanzania reported that infection was the most common VMMC AE [16]. At the 4 weeks follow-up visit, our study showed that one participant experienced delayed wound healing. In comparison, a South African prospective cohort study reported mostly infection, and some swelling accounted for the bulk of the AE experienced [14]. Variation in results among studies may be due to differences in schedules used for postoperative follow-up visits, and differences in AE reporting guidelines. Furthermore, our study showed that there was no significant association between AE and those who attended at least one follow-up visit.

Lastly, our study's VMMC AE from surgical techniques when compared with AE from medical devices such as prepex was slightly higher by 1.01%. This is because a 2016 prospective cohort pilot study done in three countries (Mozambique, South Africa, and Zambia) showed

that prevalence of AE after use of prepex was 2.0% in Zambia [21]. Furthermore, a 2016 local randomised controlled trial showed that the prevalence of AE after shang ring use was much lower at 0.6%. Differences in AE among VMMC methods are multifactorial and may include short procedure time, less surgical skill needed, and improved safety profile [22, 23]. Despite the lower prevalence of AE with medical devices, the majority of VMMC in Zambia are still via conventional surgery not device based. This is because of the gradual transition from research findings to clinical implementation, lack of standard operating procedures for shang ring till May 2020 [24] and higher overall costs [22, 23].

At the national level, our study contributes to Zambia's Ministry of Health legacy goal number 6, which aims to achieve HIV epidemic control by reducing the HIV incidence from 48,000 to 5,000. This study also provides high priority national and regional information on VMMC policy monitoring and evaluation with regards to VMMC quality (AE), surgical efficiency (VMMC surgical methods), and human resource (VMMC providers). At both national and regional level, this study confirms that the quality of the VMMC program in Ndola city, Copperbelt Province of Zambia is adequate owing to the high safety profile as evidenced by the low prevalence of VMMC AEs below 5%. At a global level, this study contributes to sustainable development goal number 3, which aims to end the AIDS epidemic by 2030.

This study had a few potential limitations. The minor change in reporting format on the VMMC client record form under demographics (i.e., marital status) occurred in the second part of 2015 could have led to bias; however, this was accounted for during coding in SPSS. This study was conducted from one tertiary hospital to represent the provincial level; hence results should be cautiously interpreted and generalised at the country level. There was incomplete or missing information in the VMMC client record form which was handled during the statistical analysis. Overall the use of secondary data for research limits the researcher as the data is not meant for research purposes.

## Conclusion

The low prevalence of AE guarantees the safety of the VMMC program in Ndola city, Copperbelt Province of Zambia. The VMMC volume was significantly associated with AEs. Therefore, adult circumcisions were safer when VMMC providers regularly performed more circumcisions (thus gained more surgical proficiency) than those who did not. Considering alternative factors affecting VMMC coverage there is also a need to explore other national and high priority areas of VMMC program quality such as adherence to follow-up visits client satisfaction, and counselling services.

## Supporting information

**S1 Data.**
(SAV)

## Acknowledgments

We wish to acknowledge the invaluable support received from the Ndola Teaching Hospital Management and specifically the Male Circumcision providers for their technical support during the study. We thank Dr Gavin George and the entire management at the school of health sciences, Kwazulu-Natal University for making this work possible.

## Author Contributions

**Conceptualization:** Imukusi Mutanekelwa.

**Data curation:** Elijah Kabelenga.

**Formal analysis:** Imukusi Mutanekelwa, Seter Siziya, Victor Daka, Elijah Kabelenga, Christopher Nyirenda, Steward Mudenda.

**Investigation:** Imukusi Mutanekelwa, David Mulenga, Steward Mudenda, Bright Mukanga, Kasonde Bowa.

**Methodology:** Imukusi Mutanekelwa, Seter Siziya, Victor Daka, Elijah Kabelenga, Ruth L. Mfune, Misheck Chileshe, David Mulenga, Herbert Tato Nyirenda, Christopher Nyirenda, Bright Mukanga, Kasonde Bowa.

**Project administration:** Imukusi Mutanekelwa.

**Supervision:** Kasonde Bowa.

**Writing – original draft:** Imukusi Mutanekelwa, Kasonde Bowa.

**Writing – review & editing:** Seter Siziya, Victor Daka, Elijah Kabelenga, Ruth L. Mfune, Misheck Chileshe, David Mulenga, Herbert Tato Nyirenda, Christopher Nyirenda, Steward Mudenda, Bright Mukanga, Kasonde Bowa.

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
