## [Decision Letter · Decision Letter 0]

8 Mar 2021

PONE-D-20-40949

Prevalence and correlates of voluntary medical male circumcision adverse events among adult males in the Copperbelt Province of Zambia: a cross-sectional study

PLOS ONE

Dear Dr. Daka,

Thank you for submitting your manuscript to PLOS ONE. After careful consideration, we feel that it has merit but does not fully meet PLOS ONE’s publication criteria as it currently stands. Therefore, we invite you to submit a revised version of the manuscript that addresses the points raised during the review process.

We look forward to receiving your revised manuscript.

Kind regards,

Richard Kao Lee, M.D.

Academic Editor

PLOS ONE

Journal Requirements:

3.We note that you have indicated that data from this study are available upon request. PLOS only allows data to be available upon request if there are legal or ethical restrictions on sharing data publicly. For information on unacceptable data access restrictions, please see http://journals.plos.org/plosone/s/data-availability#loc-unacceptable-data-access-restrictions.

Additional Editor Comments:

No major new findings outside of surgical volume-outcomes relationship which has been previously described

Reviewers' comments:

Reviewer's Responses to Questions

**Comments to the Author**

1. Is the manuscript technically sound, and do the data support the conclusions?

Reviewer #1: Yes

Reviewer #2: Partly

Reviewer #3: Yes

2. Has the statistical analysis been performed appropriately and rigorously? 

Reviewer #1: Yes

Reviewer #2: I Don't Know

Reviewer #3: Yes

3. Have the authors made all data underlying the findings in their manuscript fully available?

Reviewer #1: Yes

Reviewer #2: Yes

Reviewer #3: Yes

4. Is the manuscript presented in an intelligible fashion and written in standard English?

Reviewer #1: Yes

Reviewer #2: Yes

Reviewer #3: Yes

5. Review Comments to the Author

Reviewer #1: The authors performed a descriptive study of the AEs that were seen in a VMMC program in Ndola city of Zambia. Please see my comments below:

Introduction –

1. Line 66: key intervention for what? Please delete or edit this sentence.

2. Introduction is too long with multiple points that should be in discussion. The authors should provide a short description of the current problems of VMMC in Africa and then mention why their study is of importance. Most of the introduction points should be moved to discussion.

Methods –

1. Authors should report patient comorbidities that could be a reason of AEs

2. The MV analysis should be performed including patient comorbidities, VMMC type and timing of AE.

Discussion –

1. The authors should discuss how their AEs compare to studies using medical devices (Mogen clamp, ShangRing etc). WHO has approved many of these devices to promote circumcisions in Africa so a comparison should be made.

Reviewer #2: Reviewer Recommendation and Comments for Manuscript Number PONE-D-20-40949

Prevalence and correlates of voluntary medical male circumcision adverse events among adult males in the Copperbelt Province of Zambia: a cross-sectional study

Comments to author:

The authors have performed a cross-sectional study of prevalence of VMMC AE and their associated factors among adult males in Ndola, Zambia. This type of study is important to inform program quality and safety. The authors found a low prevalence of AEs, providing assurance that VMMC’s conducted at the study site are safe.

I am concerned as you will see in the comments below about the conclusion related to volume of procedures conducted and occurrence of AEs. While I don’t take issue with the fact that it is likely (and there are data from other’s studies showing this) that providers who conducted larger numbers of procedures had fewer AEs, I don’t think you can make this conclusion based on how the data are presented. The issue is that you have not looked at the numbers of AEs resulting from VMMCs conducted by INDIVIDUAL providers, but rather by TYPES of providers. There is no indication of how many providers are in each of the three groups or how many procedures any given provide performed. The results could be heavily biased if say one or two providers performed a large number of all the procedures or if most of the AEs occurred following VMMCs by one or two providers. If you can present results showing how many providers are in each group, the range of VMMCs performed by providers in each group and the range of AEs/provider that may make the conclusion more appropriate. I also question the inclusion of the medical offices who only conducted 7 VMMCs and the analyses looking at sleeve resection given that there were only 2 cases.

The authors should address the comments below and submit a revised version of the manuscript before the journal makes a decision. My comments are listed by line number in the PDF version of the manuscript.

Lines 75 & 76. Could the authors please clarify the statement about circumcision in Northern Africa. The term VMMC has generally been applied to the male circumcision services provided in medical settings related to HIV prevention (in fact it was not even in use to my knowledge until the establishment of the male circumcision interventions in sub-Saharan Africa beginning in 2008-2009). It thus seems inaccurate to say that VMMC is practiced extensively in Northern Africa. Are the authors referring to common occurrence of traditional circumcision for cultural or religious reasons, for example? Please clarify.

Lines 84 85. Please add that the AE rates being quotes are from various countries in sub-Saharan Africa.

Lines 101-102. The authors indicate that one of the main WHO monitoring and evaluation parameters is the number or percentage of circumcised males experiencing and adverse event during the VMMC, but I think they mean during and after.

Line 127. In the study design section, I think it would be useful to the reader to include what type of VMMC procedure(s) are used at the facility and also what follow-up schedule is used.

Line 132. Please add some additional text that explains what adopting a “campaign VMMC model” means.

Line 133. Please clarify what the seven core VMMC responsibilities are. I don’t think the average reader will understand what this means.

Line 166. Could the authors provide the reason that an ERC from South Africa was used for the study in Zambia. All of the authors of the paper are from Zambian institutions, so it seems unusual and would be good to indicate the reason.

Line 183-185. Please clarify if the exclusion was just for the variable(s) that was missing as opposed to excluding the entire form/participant.

Line 186. It might be clearer if the first section in the results is titled Adverse Events. That section could include the sentences that are on line 188-194 as that information doesn’t relate to socio-demographic characteristics.

Line 198. Please add the percentage to match the way other results are reported. Also, to reduce any confusion it might be better to say 50 + years since that is the way the results are presented in the table.

Line 201. Please add that the partial protection being referred to here (and add to the table as well) is for HIV.

Lines 201-203. It is misleading to say that the AEs were higher in those undergoing VMMC for partial protection when in fact the difference was not significant. The numerical difference is small and also the number in the “other” group is small.

Line 214 & 222. I think it would be clearer to talk about volume/cadre as opposed to just volume.

Line 217. Please add the percentage.

Lines 228 & 229. The abbreviations are confusing and don't match what is in the table. Is a registered midwife, considered a registered nurse? If not then in the table and text it would be better to have Registered nurse/midwives.

Table 2.

• Says Date of operation via Quarter, but the data are for half the year, not by quarters. The term semester is used in the text.

• Also, please check the *. I think that in two cases there is one, when there should be two (type of VMMC and follow-up visit immediately post op).

• “Follow-up visit immediately post op” Please remove the word visit assuming this is during the post-op period before they have left the facility it is confusing to use the word visit.

• I am afraid I don’t understand what the yes and no rows for the follow-up visit variables mean. If it means yes they came for a visit and no they didn’t, then how do you know if they had an AE if they didn’t come for that visit? Somehow this needs to be clarified. The same issue applies to table 3.

Lines 236-238. I find the issue of volume as related to likelihood of AEs to be confusing and I am not sure how to interpret the results. When I read the abstract, I was under the impressions that volume related to an individual provider, i.e. those individuals who had performed more VMMC, had fewer AEs. The text in the abstract is “VMMC volume (as measured by the number of surgeries conducted per VMMC provider”), which I think suggests volume relates to an individual provider. That same text appears here on line 237 & 238 (and 286 & 287). However, from looking at the data in Table 2, it seems that volume is related to a TYPE of provider (i.e., Enrolled Nurses vs. Registered Nurses vs. Medical Officers) and not an INDIVIDUAL provider as suggested by the text in the abstract, results, and discussion.

I am not sure how one interprets the results when they are presented this way because we don’t know how many providers there were in each of the type of providers (how many enrolled nurses, how many registered nurses, how many medical officers), how many VMMCs were done by each provider, and only 1.8% of all the VMMCs were done by medical officers. Maybe, for example, the medical officer was called in to do difficult cases.

Lines 242-244. I am not a statistician, but it seems questionable to me, irrespective of what the statistical testing shows, to make statement about the effect of VMMC technique on occurrence of AE’s when there were only two cases of sleeve resection

Lines 273-277. There is nothing in the results to indicate what VMMC procedure was done in the 2 AEs that occurred during the operation, but based on the text here it seems that the 1 AE reported intra-op was in a sleeve resection. I think it would be useful to put this information in the results. I wonder about the conclusion that this slightly higher rates of intraop AE was “probably attributed to VMMC providers with less surgical proficiency and the use of surgically advanced VMMC methods (i.e. Sleeve Resection) in less experienced clinician hands”. Is there some information to substantiate that the providers who did the sleeve resections were less experienced or less surgically proficient?”

Lines 292-295. “This study did not focus on the quality of surgical techniques necessary to compare AEs among the VMMC providers using a prospective cohort study. Hence, AEs differences among VMMC providers is probably due to differences in VMMC volume.” I am not sure what the first sentence means here and also that just because you didn’t focus on the quality doesn’t mean the difference isn’t due to the quality.

Lines 299-301. I don’t think it is appropriate to make the following statement: “Despite all VMMC providers in our study being certified surgically competent, few were surgically proficient as evidenced by the frequency and number of VMMC conducted.” There are no data presented on frequency or number provided by individual providers on which to base that statement.

Lines 301-307. Again, the conclusions don’t really follow from the data. The study in Kenya that is quoted look at numbers of VMMC conducted by individual providers, not types of providers and so it was appropriate for them to conclude that those conducting more procedures had fewer AEs. Here, however, without knowing how many providers there are in each of the three groups or how many procedures different providers performed, I don’t believe you can make these conclusions. Perhaps there were one or two provider who performed most of the procedures in one of the nurse group who were either more or less skilled and perhaps a less skilled provider was responsible for most of the AES. Or as noted above, perhaps the medical officers were called for difficult cases and that is why it seems their AE rate was high.

Lines 341-343. I don’t recall any mention of a change in the form in the methods. Please clarify there or here what that change was.

For Figure 2, please indicate how many men/records there were for the follow-up visits at the three time points. The reader will wonder if all men had data for all of the follow-up visits.

Reviewer #3: Excellent and well thought out manuscript with rigorously executed methodology. Good addition to the literature on safety of VMMC in Sub-Saharan Africa. I would only suggest the authors consider describing AE severity distributions (total and by providers). Authors should also consider reporting how the degree of severity was determined (individual provider? consensus?) The manuscript may also benefit from a statement providing more details on the post-operative AE collection mechanisms: did patient have the possibility to call their providers re: AEs outside of study visits? If so, were those calls systematically captured using one of the collected forms? Also, while AEs beyond six weeks post VMMC have been rare, how long after the procedure would a patient's report of VMMC-related AE would have been recorded? Beyond this minor modifications, I look forward to seeing this manuscript published.

6. PLOS authors have the option to publish the peer review history of their article (what does this mean?). If published, this will include your full peer review and any attached files.

Reviewer #1: No

Reviewer #2: No

Reviewer #3: **Yes: **Quincy Nang, MD, MPH

---

## [Author Response · Author response to Decision Letter 0]

14 Jun 2021

RESPONSE TO REVIEWERS 

Reviewer #1: The authors performed a descriptive study of the AEs that were seen in a VMMC program in Ndola city of Zambia. Please see my comments below:

Introduction –

1. Line 66: key intervention for what? Please delete or edit this sentence.

Response: We have revised as suggested 

2. Introduction is too long with multiple points that should be in discussion. The authors should provide a short description of the current problems of VMMC in Africa and then mention why their study is of importance. Most of the introduction points should be moved to discussion.

Response: We revised as suggested. 

Methods –

1. Authors should report patient comorbidities that could be a reason of AEs

2. The MV analysis should be performed including patient comorbidities, VMMC type and timing of AE.

Response: Noted, these data were not available for analysis and therefore was not included in the MV analysis. 

Discussion –

1. The authors should discuss how their AEs compare to studies using medical devices (Mogen clamp, ShangRing etc.). WHO has approved many of these devices to promote circumcisions in Africa so a comparison should be made.

Response: We revised as suggested 

Reviewer #2: Reviewer Recommendation and Comments for Manuscript Number PONE-D-20-40949

Prevalence and correlates of voluntary medical male circumcision adverse events among adult males in the Copperbelt Province of Zambia: a cross-sectional study

Comments to author:

The authors have performed a cross-sectional study of prevalence of VMMC AE and their associated factors among adult males in Ndola, Zambia. This type of study is important to inform program quality and safety. The authors found a low prevalence of AEs, providing assurance that VMMC’s conducted at the study site are safe.

I am concerned as you will see in the comments below about the conclusion related to volume of procedures conducted and occurrence of AEs. While I don’t take issue with the fact that it is likely (and there are data from other’s studies showing this) that providers who conducted larger numbers of procedures had fewer AEs, I don’t think you can make this conclusion based on how the data are presented. The issue is that you have not looked at the numbers of AEs resulting from VMMCs conducted by INDIVIDUAL providers, but rather by TYPES of providers. There is no indication of how many providers are in each of the three groups or how many procedures any given provide performed. The results could be heavily biased if say one or two providers performed a large number of all the procedures or if most of the AEs occurred following VMMCs by one or two providers. If you can present results showing how many providers are in each group, the range of VMMCs performed by providers in each group and the range of AEs/provider that may make the conclusion more appropriate. I also question the inclusion of the medical offices who only conducted 7 VMMCs and the analyses looking at sleeve resection given that there were only 2 cases.

The authors should address the comments below and submit a revised version of the manuscript before the journal makes a decision. My comments are listed by line number in the PDF version of the manuscript.

Lines 75 & 76. Could the authors please clarify the statement about circumcision in Northern Africa. The term VMMC has generally been applied to the male circumcision services provided in medical settings related to HIV prevention (in fact it was not even in use to my knowledge until the establishment of the male circumcision interventions in sub-Saharan Africa beginning in 2008-2009). It thus seems inaccurate to say that VMMC is practiced extensively in Northern Africa. Are the authors referring to common occurrence of traditional circumcision for cultural or religious reasons, for example? Please clarify.

Response: Noted, we revised as suggested

Lines 84 85. Please add that the AE rates being quotes are from various countries in sub-Saharan Africa. 

Response: We revised as suggested

Lines 101-102. The authors indicate that one of the main WHO monitoring and evaluation parameters is the number or percentage of circumcised males experiencing and adverse event during the VMMC, but I think they mean during and after.

Response: Noted, we revised as suggested

Line 127. In the study design section, I think it would be useful to the reader to include what type of VMMC procedure(s) are used at the facility and also what follow-up schedule is used.

Response: We revised as suggested

Line 132. Please add some additional text that explains what adopting a “campaign VMMC model” means.

Response: Addition made as suggested

Line 133. Please clarify what the seven core VMMC responsibilities are. I don’t think the average reader will understand what this means.

Response: Clarification made as suggested 

Line 166. Could the authors provide the reason that an ERC from South Africa was used for the study in Zambia. All of the authors of the paper are from Zambian institutions, so it seems unusual and would be good to indicate the reason.

Response: At the time of the study the principal and first author was based and affiliated to University of KwaZulu Natal. 

Line 183-185. Please clarify if the exclusion was just for the variable(s) that was missing as opposed to excluding the entire form/participant.

Response: We revised as advised

Line 186. It might be clearer if the first section in the results is titled Adverse Events. That section could include the sentences that are on line 188-194 as that information doesn’t relate to socio-demographic characteristics.

Response: We revised as advised

Line 198. Please add the percentage to match the way other results are reported. Also, to reduce any confusion it might be better to say 50 + years since that is the way the results are presented in the table.

Response: We revised as advised

Line 201. Please add that the partial protection being referred to here (and add to the table as well) is for HIV.

Response: We revised as advised

Lines 201-203. It is misleading to say that the AEs were higher in those undergoing VMMC for partial protection when in fact the difference was not significant. The numerical difference is small and also the number in the “other” group is small.

Response: We revised as advised

Line 214 & 222. I think it would be clearer to talk about volume/cadre as opposed to just volume.

Response: Noted, with the advent of task shifting, the study focus was on volume. Furthermore 98.2% (384/391) of VMMC was performed by nurses 

Line 217. Please add the percentage.

Response: Added as advised

Lines 228 & 229. The abbreviations are confusing and don't match what is in the table. Is a registered midwife, considered a registered nurse? If not then in the table and text it would be better to have Registered nurse/midwives.

Response: Adjusted as advised, registered nurses (i.e. diploma holders) can have no specialty i.e. registered general nurse or with theater specialty i.e. registered theater nurse or with obstetric specialty i.e. registered midwife

Table 2.

• Says Date of operation via Quarter, but the data are for half the year, not by quarters. The term semester is used in the text.

• Also, please check the *. I think that in two cases there is one, when there should be two (type of VMMC and follow-up visit immediately post op).

• “Follow-up visit immediately post op” Please remove the word visit assuming this is during the post-op period before they have left the facility it is confusing to use the word visit.

• I am afraid I don’t understand what the yes and no rows for the follow-up visit variables mean. If it means yes they came for a visit and no they didn’t, then how do you know if they had an AE if they didn’t come for that visit? Somehow this needs to be clarified. The same issue applies to table 3.

Response: Revised as advised. No means absconded follow-up visit. Retrospective data was used, adverse effects in the those who didn’t attend follow-up visits were not assessed and handled as missing variables in the statistical analysis. 

Lines 236-238. I find the issue of volume as related to likelihood of AEs to be confusing and I am not sure how to interpret the results. When I read the abstract, I was under the impressions that volume related to an individual provider, i.e. those individuals who had performed more VMMC, had fewer AEs. The text in the abstract is “VMMC volume (as measured by the number of surgeries conducted per VMMC provider”), which I think suggests volume relates to an individual provider. That same text appears here on line 237 & 238 (and 286 & 287). However, from looking at the data in Table 2, it seems that volume is related to a TYPE of provider (i.e., Enrolled Nurses vs. Registered Nurses vs. Medical Officers) and not an INDIVIDUAL provider as suggested by the text in the abstract, results, and discussion.

I am not sure how one interprets the results when they are presented this way because we don’t know how many providers there were in each of the type of providers (how many enrolled nurses, how many registered nurses, how many medical officers), how many VMMCs were done by each provider, and only 1.8% of all the VMMCs were done by medical officers. Maybe, for example, the medical officer was called in to do difficult cases.

Response: Table 2 shows that enrolled nurses performed 63.2% (247/391) VMMC’s, registered nurses 35.0% (137/391) and medical officers 1.8% (7/391)

Lines 242-244. I am not a statistician, but it seems questionable to me, irrespective of what the statistical testing shows, to make statement about the effect of VMMC technique on occurrence of AE’s when there were only two cases of sleeve resection

Response: Noted, however, there is also a disclaimer that the “result must be interpreted with extreme caution because of the few numbers of Sleeve Resections done”

Lines 273-277. There is nothing in the results to indicate what VMMC procedure was done in the 2 AEs that occurred during the operation, but based on the text here it seems that the 1 AE reported intra-op was in a sleeve resection. I think it would be useful to put this information in the results. I wonder about the conclusion that this slightly higher rates of intraop AE was “probably attributed to VMMC providers with less surgical proficiency and the use of surgically advanced VMMC methods (i.e. Sleeve Resection) in less experienced clinician hands”. Is there some information to substantiate that the providers who did the sleeve resections were less experienced or less surgically proficient?”

Response: We reviewed as advised, a Junior Resident Medical Officer (also called an intern doctor) performed the sleeve resection.

Lines 292-295. “This study did not focus on the quality of surgical techniques necessary to compare AEs among the VMMC providers using a prospective cohort study. Hence, AEs differences among VMMC providers is probably due to differences in VMMC volume.” I am not sure what the first sentence means here and also that just because you didn’t focus on the quality doesn’t mean the difference isn’t due to the quality.

Response: we revised as advised

Lines 299-301. I don’t think it is appropriate to make the following statement: “Despite all VMMC providers in our study being certified surgically competent, few were surgically proficient as evidenced by the frequency and number of VMMC conducted.” There are no data presented on frequency or number provided by individual providers on which to base that statement.

Response: We revised as advised

Lines 301-307. Again, the conclusions don’t really follow from the data. The study in Kenya that is quoted look at numbers of VMMC conducted by individual providers, not types of providers and so it was appropriate for them to conclude that those conducting more procedures had fewer AEs. Here, however, without knowing how many providers there are in each of the three groups or how many procedures different providers performed, I don’t believe you can make these conclusions. Perhaps there were one or two provider who performed most of the procedures in one of the nurse group who were either more or less skilled and perhaps a less skilled provider was responsible for most of the AES. Or as noted above, perhaps the medical officers were called for difficult cases and that is why it seems their AE rate was high.

Response: Revised as advised under “VMMC operation details” first paragraph. VMMC’s were done by 7 individual providers throughout the year 2015. More than half (63.2%) of them were by two individuals enrolled nurses, 35% by three individuals registered nurses and 1.8% by two individual medical officers

Lines 341-343. I don’t recall any mention of a change in the form in the methods. Please clarify there or here what that change was.

Response: We revised as advised

For Figure 2, please indicate how many men/records there were for the follow-up visits at the three time points. The reader will wonder if all men had data for all of the follow-up visits.

Response: The information of total men/records is also provided in table 2 to avoid overcrowding of data

Reviewer #3: Excellent and well thought out manuscript with rigorously executed methodology. Good addition to the literature on safety of VMMC in Sub-Saharan Africa. I would only suggest the authors consider describing AE severity distributions (total and by providers). Authors should also consider reporting how the degree of severity was determined (individual provider? consensus?) The manuscript may also benefit from a statement providing more details on the post-operative AE collection mechanisms: did patient have the possibility to call their providers re: AEs outside of study visits? If so, were those calls systematically captured using one of the collected forms? Also, while AEs beyond six weeks post VMMC have been rare, how long after the procedure would a patient's report of VMMC-related AE would have been recorded? Beyond this minor modifications, I look forward to seeing this manuscript published.

Response: Noted, data on AE severity were not available for analysis. Retrospective data from hospital records were used and there was no evidence of active follow-up of post-operative AE using phone calls. Individuals with AE who presented themselves to the hospital had the AE’s recorded

---

## [Decision Letter · Decision Letter 1]

20 Aug 2021

Prevalence and correlates of voluntary medical male circumcision adverse events among adult males in the Copperbelt Province of Zambia: a cross-sectional study

PONE-D-20-40949R1

Dear Dr. Daka,

We’re pleased to inform you that your manuscript has been judged scientifically suitable for publication and will be formally accepted for publication once it meets all outstanding technical requirements.

Kind regards,

Richard Kao Lee, M.D.

Academic Editor

PLOS ONE

Additional Editor Comments (optional):

Reviewers' comments:

Reviewer's Responses to Questions

**Comments to the Author**

1. If the authors have adequately addressed your comments raised in a previous round of review and you feel that this manuscript is now acceptable for publication, you may indicate that here to bypass the “Comments to the Author” section, enter your conflict of interest statement in the “Confidential to Editor” section, and submit your "Accept" recommendation.

Reviewer #1: All comments have been addressed

2. Is the manuscript technically sound, and do the data support the conclusions?

Reviewer #1: Yes

3. Has the statistical analysis been performed appropriately and rigorously? 

Reviewer #1: Yes

4. Have the authors made all data underlying the findings in their manuscript fully available?

Reviewer #1: Yes

5. Is the manuscript presented in an intelligible fashion and written in standard English?

Reviewer #1: Yes

6. Review Comments to the Author

Reviewer #1: The authors performed a descriptive study of the AEs that were seen in a VMMC program in Ndola city of Zambia. The authors have adequately addressed all the comments.

7. PLOS authors have the option to publish the peer review history of their article (what does this mean?). If published, this will include your full peer review and any attached files.

Reviewer #1: No

---

## [Editor Report · Acceptance letter]

27 Aug 2021

PONE-D-20-40949R1 

Prevalence and correlates of voluntary medical male circumcision adverse events among adult males in the Copperbelt Province of Zambia: a cross-sectional study 

Dear Dr. Daka:

I'm pleased to inform you that your manuscript has been deemed suitable for publication in PLOS ONE. Congratulations! Your manuscript is now with our production department. 

Kind regards, 

on behalf of

Dr. Richard Kao Lee 

Academic Editor

PLOS ONE